# Cool-Clave—An Energy Efficient Autoclave

**Indraneel R. Chowdhury** 🆔 and **John Summerscales** *🆔

School of Engineering, Computing and Mathematics, University of Plymouth, Drake Circus,
Plymouth PL4 8AA, UK
* Correspondence: j.summerscales@plymouth.ac.uk

**Abstract:** Out-of-autoclave (OOA) manufacturing techniques for composites result in lower fibre volume fractions than for fully compressed laminates. The lower fibre volume fraction produces a higher resin volume fraction, which becomes resin-rich volumes (RRV). Textile reinforcements with clustered fibres and consequent RRV generally have low strength but high in-plane process permeability, whereas the opposite is true for uniformly distributed fibres. The inevitable increase in resin volume fraction of OOA composites often compromises composite performance and leads to relatively higher weight and fuel consumption in transport applications. The retention of autoclave processing is recommended for highest performance when compression press moulding is not appropriate (for example, for complex 3D components). The traditional autoclave processing of composites heats not only the component to be cured but also parasitic air and the vessel insulation. Subject to minor modifications of the pressure vessel, electrically heated tooling could be implemented. This approach would need to balance insulation of the heated tool surface (and any heater blanket on the counter-face) against the quenching effect during the introduction of the pressurised cool air. This process optimisation would significantly reduce energy consumption. Additionally, the laminate on the heated tool could be taken to the end of the dwell period before loading the autoclave, leading to significant reductions in cure cycle times. Components could be cured simultaneously at different temperatures provided that there are sufficient power and control circuits in the autoclave. While autoclave processing has usually involved vacuum-bagged pre-impregnated reinforcements, implementation of the cool-clave technique could also provide a scope for using the pressure vessel to cure vacuum-infused composites.

**Keywords:** autoclave; cool-clave; vacuum; heated tooling; fibre-reinforced composites





## 1. Introduction

Composite materials are widely used in industrial applications due to their unique characteristics of high-stiffness-to-weight ratio, excellent durability, chemical resistance, and better recycling potential as compared to metallic components. High performance composites, mainly used in aerospace applications, are produced in the autoclave by applying elevated pressure and temperature [1,2]. However, autoclave processing of composites involves long curing cycle times, expensive tooling, and high energy consumption [3,4]. As a result, there is an interest across the range of composites manufacturing processes for cost reduction with a current focus on out-of-autoclave (OOA) processes [1,5], especially OOA prepreg [6] and resin infusion under flexible tooling [7–9]. The OOA process involves manufacturing composites by applying vacuum and heat outside of the autoclave, but has limitations on the maximum laminate fibre volume fraction due to compressibility characteristics of the reinforcement. As a result, composites manufactured by vacuum-only processes cannot achieve high fibre volume contents, which is a primary requirement in high performance composites for aerospace, automobile, and defence sectors. Compression moulding in a hydraulic press creates limited compaction perpendicular to the line of action of the press. The autoclave is the best process for consolidation of complex three-dimensional components, but suffers from several limitations: (a) pre-impregnated

(prepreg) reinforcements incur a high-cost premium, (b) high energy input for heating and air circulation: flow speeds in the range 1.4–2.1 m/s [5], (c) non-uniform heating of the components in the vessel due to turbulent flow, windward vs. leeward location, flow stagnation and consequent temperature differences, (d) thermal lag due to the tool or consumables between the heat source and the composite, and (e) long cycle times, which may be a "bottleneck" constraint [10,11].

However, with minor modifications of the pressure vessel, there is potential to improve the efficiency of autoclave processes. The traditional autoclave not only heats the composite parts to be cured but also inert gasses and vessel insulation. Significant energy savings can result from using electrically heated tooling to only heat the essential parts of the process (the tool and composite), and by cool-air pressurisation of composites [12]. Further to the potential for significant reductions in energy consumption, the laminate on the heated tool could be taken to the end of the dwell period before loading the autoclave, leading to significant reductions in cycle times. Autoclave loading efficiency could be improved by curing different composite systems simultaneously with the composites brought to their respective curing temperatures before loading the autoclave, provided there are sufficient power and control circuits in the autoclave, which would further enhance process efficiency [12].

This paper critically reviews the technique of using heated tooling in the autoclave to enhance the energy- and cost-efficiency of autoclave process, designated as the 'cool-clave' technique.

## 2. Autoclave

Autoclaves have become indispensable equipment to process high-quality polymer composite materials for structural industries, such as aerospace, automotive, and defence sectors [13]. Today, for example, in the aircraft industry, investments in such equipment are strategically important. Autoclaves are now being used to produce very large aircraft components, such as wing and fuselage sections. They can process a wide variety of materials, including thermoset [14] and thermoplastic [15]-based composite parts with varying contours and complex shapes.

The quality requirements of the present high-performance composites for aerospace/ defence industries are indeed more stringent. Additionally, there is an urgent requirement to improve the efficiency and cost-effectiveness of high-performance structural composites, along with ensuring reliable and consistent processing methods. Therefore, it is imperative for autoclave design engineers to take into consideration different governing criteria to address the diverse and complex requirements for developing state-of-the-art autoclave systems. In addition to handling a wide variety of consumables, modern autoclaves must respect health and safety requirements [16] and ensure minimum maintenance costs.

Autoclaves are closed pressure vessels used to manufacture high performance composite components. Uncured composites are moulded in an autoclave typically heated using inert gases, such as carbon dioxide or nitrogen, thus allowing the transfer of heat and pressure to the composite component for consolidation and allowing it to cure firmly and uniformly. The application of pressure for consolidation of composites in an autoclave helps in reducing porosity and voids, retains shape around the mould, and enables better control to maintain a higher fibre volume fraction in composite components [17]. The autoclave process draws many similarities with hot pressing technique; however, the main difference pertains to the way heat and pressure are applied [18]. The autoclave operating parameters, such as temperature and pressure are based on the resin systems used. Generally, epoxy resins require temperatures < 200 °C and pressures of 0.7 MPa [17].

Figure 1 shows the internal chamber of the Aeroform autoclave located in the composites manufacturing laboratory at the University of Plymouth. Surrounding the main internal area is a metal inner case which shields the components being cured from twelve electric heating elements positioned at intervals around the chamber's circumference. Behind the heating elements is a layer of thermal insulation, protected by sheet metal. As the autoclave

walls are made up of quality carbon steel, up to 150 mm on some autoclaves [17,18], they act as a heat sink. The insulation is designed to minimise heat transfer from the main chamber to the autoclave walls. The insulation commonly used consists of refractory ceramics or fibreglass insulation. The insulation prevents excess energy loss and is designed to keep the autoclave outer walls down to a maximum temperature of 60 °C.

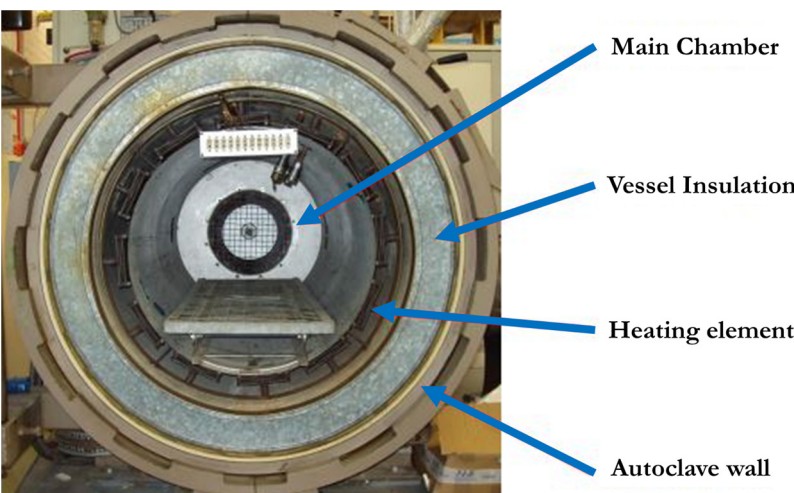

**Figure 1.** Internal chamber of 0.67 m working diameter Aeroform autoclave at the University of Plymouth.

The design of autoclave systems is multidisciplinary in nature and encompasses mechanical, process control, and instrumentation engineering. Invariably, the state-of-the-art autoclave systems are completely computer-controlled and are semi- or fully automated. The computer controls of these modern autoclaves are required to execute the selected cure cycle by sequentially starting various subsystems, download set values at regular time intervals to the front-end controllers, acquire, store, and archive the data, monitor cure status and faults, generate alarms, and perform the functions of sequential shut down and reporting [19]. Ease of maintenance, fail-safe operation, and reliability are among the key drivers in modern autoclaves. The low cost of ownership also needs to be considered in today's context [17]. In recent years, there has been an increasing demand to enhance the service temperature of high performance structural composite components, invariably leading to higher curing temperature and pressure requirements in the order of 300 to 350 °C and up to 1.5 MPa. This necessitates the development of high temperature and high-pressure autoclave systems, which presents a new set of challenges such as the handling of massive door and locking systems, temperature uniformity, special material requirements for door and shell flanges, fabrication, transportation, and, most importantly, low cost and maintenance requirements [17,19].

### 2.1. Autoclave Moulding

In an autoclave moulding, isostatic pressure is applied to the composite component on a vacuum bagged mould prior to applying heat/vacuum/pressure to compact the composite material. The intimacy of contact between the composite and mould therefore depends on the magnitude of the applied pressure. Pre-impregnated (prepreg) reinforcements cured by autoclave processing are first covered in peel-ply, with bleeders used to draw out excess air and to soak up excess resin while curing (unless the system is intended for zero-bleed), prior to vacuum bagging, as this allows for the manufacturing of composite components with a high-quality surface finish. During autoclave processing, engineers adjust the process by maximising air expulsion and minimising excess resin flow. Curing pressures typically ranges between 3–12 MPa [20,21]. Prepreg reinforcements are generally

used as they offer good processability. The fundamental components of an autoclave moulding process are shown in Figure 2.

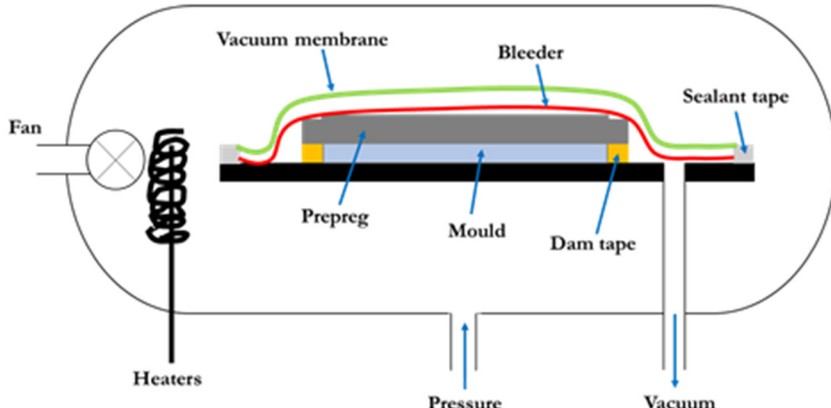

**Figure 2.** Fundamental components of an Autoclave process (redrawn from [20]).

Autoclave manufacturing is generally a 3-stage process. In the first stage, the vacuum is applied until the desired temperature is reached for composite consolidation. The ramped heating enables resin viscosity to gradually decrease, as well as allowing for the release of volatiles and air bubbles. The reduced viscosity of the resin simultaneously improves wetting of the fibres and resin flow, which in turn facilies movements of volatiles and air bubbles. In the second stage, after a suitable dwell period, the consolidation pressure is applied. The final stage raises temperature to post-cure the composite, following which the resin viscosity stabilises and the material starts to cure [20]. A typical autoclave moulding bagging mechanism is further illustrated in Figure 3.

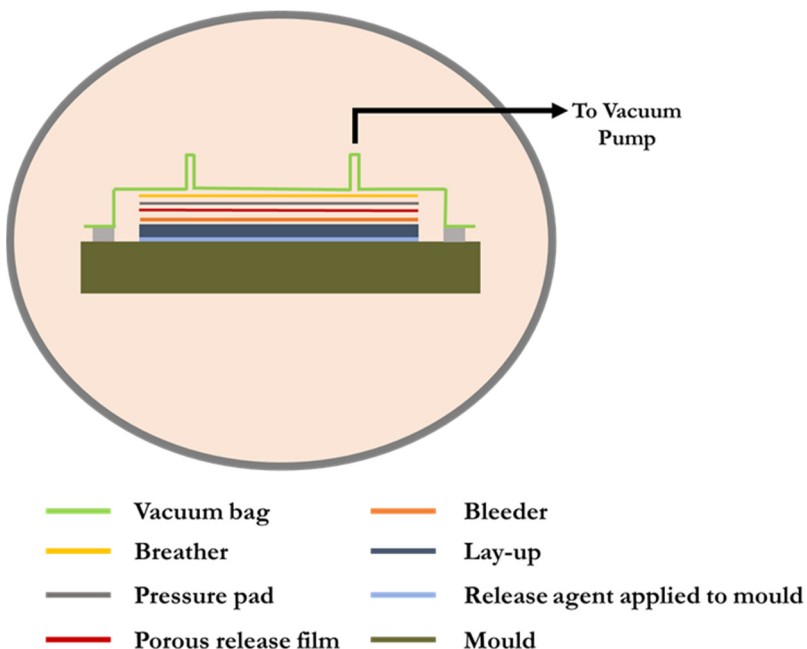

**Figure 3.** A typical autoclave moulding bagging mechanism.

In thermoset resins, curing is a chemical cross-linking process that is very crucial for optimum performance of thermoset matrix composites. For thermoplastic matrix composites, autoclave consolidation is achieved at high temperatures and bagging materials, and techniques used for thermoset composites are adapted to suit thermoplastics [22].

Autoclave moulding process comprises the following advantages [23]:

(a)  Applicable to both thermosetting and thermoplastic fibre-reinforced composites
(b)  Facilitates better inter-layer adhesion
(c)  Enables good fibre wetting
(d)  Enables higher uniformity in composite solidification
(e)  Supports superior fibre volume fraction in composite component
(f)  Reduced void content in the composite due to the vacuum/pressure
(g)  Facilitates manufacturing composite parts with high strength-to-weight ratio

The benefits of the autoclave are that the heat and pressure deliver high performance components for the composite industry, to compete with and replace metal components. Therefore, lightweight components, for example composite motor vehicle parts, can achieve higher efficiencies. In this scenario, engines will require less torque and fuel than when used to move a heavy metal structure. The disadvantages include high capital cost of the machine, and non-flexible and poor heat transfer efficiency. To heat the composite component, it is normal to parasitically heat all the air in the pressure vessel and the thermal insulation of the vessel. Most of the energy that is put into the system is taken up by the heater, therefore creating a higher consumption of energy. Other disadvantages include higher waiting times due to slow ramp rates and longer curing cycles [24].

### 2.2. Heating and Air-Circulation System in an Autoclave

Forced gas circulation systems using nitrogen or carbon dioxide are most commonly used in autoclaves. The air circulation system consists of a centrifugal blower and ducting system. The heating elements are placed around the impeller. The centrifugal blower takes in gas axially and discharges it radially to pass over the heating elements at a velocity of 1–2 m/s at ambient conditions. The air circulation system also helps in accelerating the cooling process by removing the gas from the outer surface of the cooling tubes at an increasing rate. Modern autoclaves are mechanised with a flange-mounted blower motor encased within a pressure-tight casing and connected to the rear of the autoclave. This enables the motor rotor, stator, and the mechanical components, for example, bearings to directly encounter the autoclave pressure. Power ratings of a typical autoclave range between 100 to 150 kW [17].

Heating system in autoclaves are either electrically controlled or controlled by indirect gas firing (circulating externally heated or cooled thermic fluid). However, the majority of the autoclaves are electrically heated, as these systems are cleaner and more compatible to modern computer control systems and provide better control of autoclave temperature. The electrical heating requirement in an autoclave is based on the charge and resin system requirements for the cure cycle. For example, a typical 4.5-m diameter × 9-m length autoclave requires a heating capacity of approximately 1 MW. Heating elements, typically ranging between 5–10 kW, are usually manufactured using nichrome/kanthal filament with an outer sheath of steel grouped together in banks and connected in star or delta configuration [17]. The air circulation and heating system of a typical autoclave is illustrated in Figure 4.

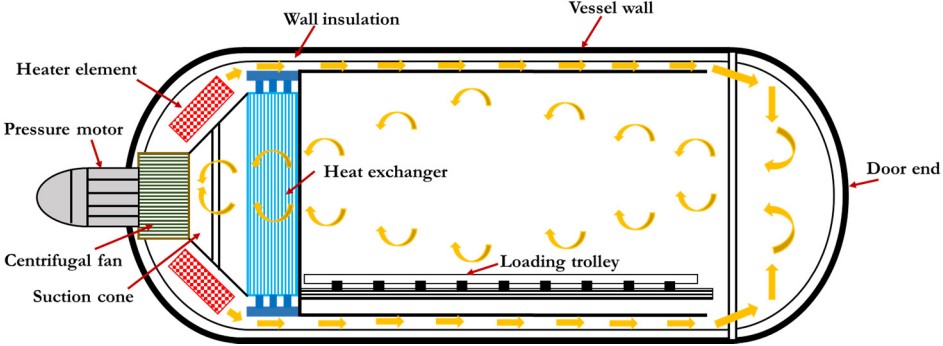

**Figure 4.** Schematic of air circulation and heating system in a typical autoclave (redrawn from [17]).

### 2.3. Pressurization and Cooling System

The pressurization systems in an autoclave ensure that the required pressurization rate is maintained during autoclave processing of composites. The average pressurization rate in modern autoclaves is 0.2 MPa/min [17]. Many modern autoclaves are nitrogen-pressurized instead of air-pressurized since autoclave cure consumables are highly inflammable in air medium due to the presence of oxygen. The pressurization system in an autoclave consists of a primary compressor, booster compressor, storage tanks, and piping circuitry. The primary compressor takes in air from the atmosphere and pressurizes it to 0.7 MPa. The booster compressor further pressurizes the air to high pressure (typically in the range between 1.7–2.2 MPa) in order to create sufficient pressure differentials to attain the required pressurization rate. In nitrogen-pressurized autoclaves, the nitrogen plant receives the air from the primary compressor at 0.7 MPa pressure and isolates nitrogen from atmospheric air by a process called Pressure Swing Adsorption (PSA) [17], which can produce nitrogen in the order of 99% purity suitable for curing polymeric composite materials. Nitrogen is then pressurized by the booster compressor to meet the required pressurization rate and purged into the chamber [17]. In Figure 5, the cooling and pressurization system used in modern autoclaves is schematically represented.

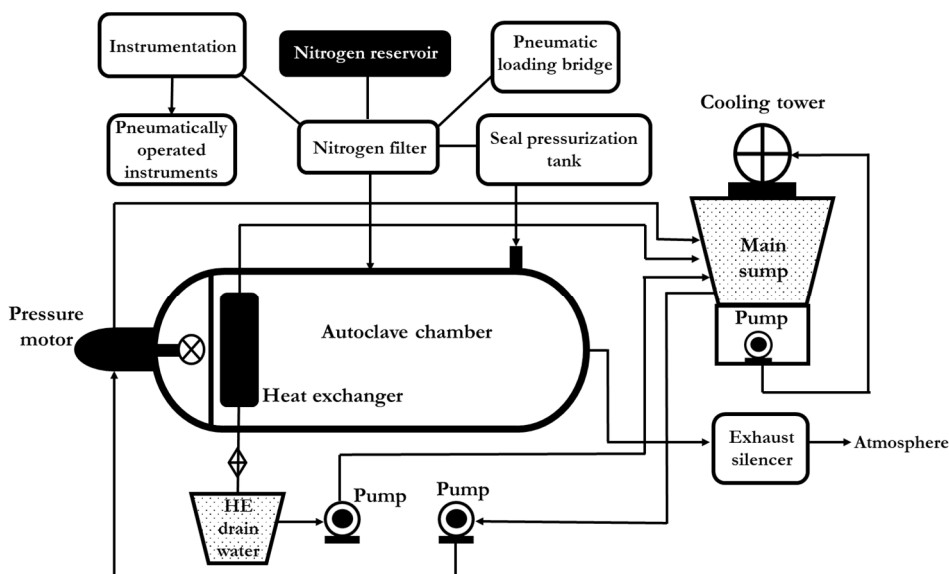

**Figure 5.** Schematic of cooling and pressurization system in an autoclave (redrawn from [17]).

Autoclave processing requires variable cooling rates based on the resin system used. Other variables affecting the cooling system are (a) temperature difference between the autoclave ambient and cooling medium, (b) flow rate of cooling medium, (c) area of heat transfer, (d) type of flow of cooling medium—parallel or cross flow, (e) conductivity of cooling coil material, and (f) velocity of autoclave medium across heat exchanger. The cooling rate in an autoclave is controlled by varying the flow of cooling fluid. In some autoclaves, both air and water are used as the cooling medium. In autoclaves, a closed loop cooling system is generally employed to prevent excess use of water. Liquid nitrogen is also used for faster cooling. A major challenge in designing cooling system for autoclaves is how fast and effectively the cooling medium can be drained from the autoclave heat exchanger, as any delay in draining the cooling medium from heat exchanger can lead to loss of heat during the heating phase and damage to the heat exchanger tubes. In the water-based cooling system of autoclaves, the simplest way of draining the water is to provide a sump just below the autoclave heat exchanger and then pump back the water to the cooling water sump. This process helps in reducing wastage of water and at the same time also prevents the steam from entering the cooling tower [17].

### 2.4. Vacuum System

Figure 6 illustrates the vacuum system of a typical autoclave. The main components of vacuum system in modern autoclaves are vacuum pumps, vacuum reservoirs, buffer tanks, suction, and measurement lines. All modern autoclaves must include an adequate number of vacuum ports and must also have the capacity to maintain different levels of vacuum in different bagging systems simultaneously. Therefore, vacuum pumps and reservoirs in an autoclave must have adequate buffer capacity. For example, vacuum pump with minimum capacity of 7 m$^3$/h is necessary for a bagging area of approximately 1 m$^2$. Generally, a 4.5 m × 9 m autoclave can have approximately 60 measurement and suction lines. Correspondingly, the pump capacity ranges to about 180 m$^3$/h with a reservoir capacity of 6 m$^3$. Vacuum requirement ranges between 667–26,664 × 10$^{-6}$ MPa based on the curing system [17].

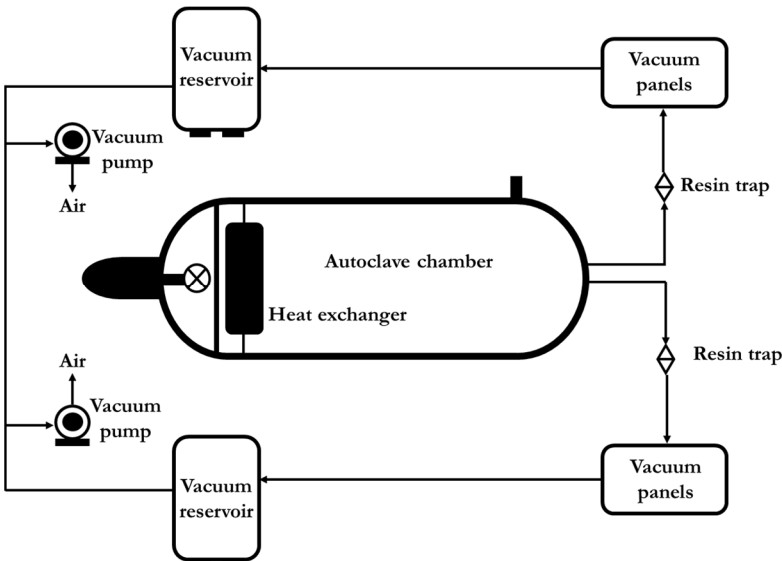

**Figure 6.** Schematic of vacuum system in a typical autoclave (redrawn from [17]).

### 2.5. Loading System

The loading system in an autoclave facilitates positioning of composite components or moulds to be cured. A loading crab was used to move the loading platform in an autoclave. Generally, autoclaves are installed in a pit with the top surface of the loading platform in flush with the floor for convenience in loading components. A loading bridge is generally deployed to bridge the gap between autoclave door and pit.

### 2.6. Electrical and Control System

Electrical and control systems in autoclaves play a key role in ensuring safe operation for reliable processing and curing of composite components. The electrical and control system of autoclaves should be robust enough to be able to provide necessary feedback signals and respond to various commands for processing [25]. Autoclaves are generally built with computer system, serial port servers, power supplies, sensors, and are generally resilient enough to be able to operate even if one or more components fail. Electrical and control systems in autoclaves are in-built with capabilities to be operated in multiple modes of operations, for examples, automatic, semi-automatic, or in manually operational mode. The control system generally consists of PID (proportional integral derivative) controllers, set temperature, and vacuum and pressure levels. PLC (programmable logic controllers) are used to ensure safe interlocking, sequential operation, status, and alarm display. All the components are generally connected to a server and computer-controlled via ethernet links. The communication system used in autoclaves includes RS485, USB, and ethernet connection [26].

### 3. Cool-Clave Technique

Autoclave processing is used to manufacture high performance composites, but overall, the process is very expensive and requires high energy consumption. In a traditional autoclave, the process not only heats up the composite to be consolidated but also any parasitic air and vessel insulation requiring high energy usage. Additionally, autoclave processing is normally used to consolidate vacuum-bagged pre-impregnated reinforcements. Prepreg materials require freezer storage, and the separate impregnation stage incurs further costs. The cool-clave technique has the potential to achieve autoclave consolidation of fibre-reinforced composites by enhancing the energy- and cost-efficiency: (a) without heating parasitic materials (hence saving energy), (b) with shorter cycle times (hence increased production rate), (c) without prepreg, i.e., using infused laminates (for lower cost). A schematic of the cool-clave technique with the bagging process is illustrated in Figure 7.

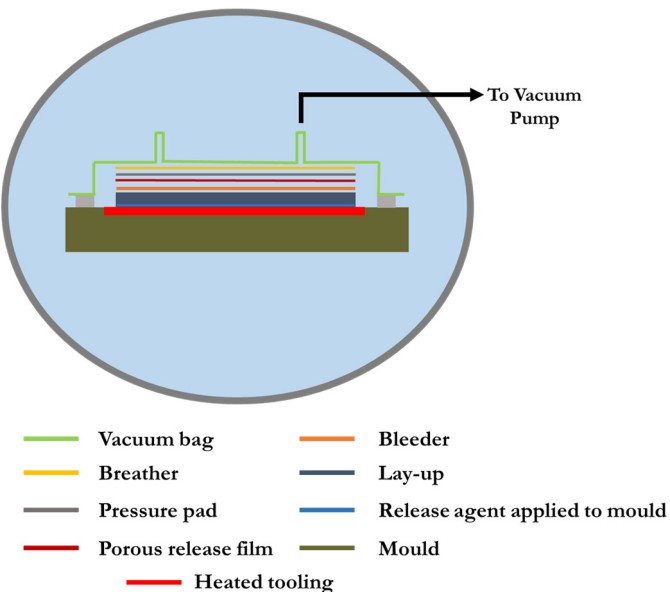

**Figure 7.** Schematic of cool-clave processing technique.

Implementation of the cool-clave technique for autoclave consolidation of composites could generate significant cost and energy savings. The equipment required for the cool-clave process consists of:

- pressure vessel (autoclave)
- heated mould tool and control unit
- efficient thermal insulation of the bagged component

Prudham and Summerscales [27] investigated the cool-clave processing technique using the autoclave at the University of Plymouth. The autoclave was used as a pressure vessel, without using the heating aspect of the system. The heat, required for curing, was provided by a mould tool with built-in heating elements. Figure 8 demonstrates the heated mould tool used for the cool-clave processing.

For implementation of the cool-clave processing technique, temperature uniformity across the face of the mould tool is crucial [18]. During the initial stages of the cure cycle, dwell periods are often used, where the laminate is held at a temperature lower than the curing temperature for a set period of time. This increase in temperature lowers the viscosity of the resin, allowing it to flow. Together with low pressure, the volatiles trapped within the laminate are forced out, thus reducing voids. If the viscosity of the resin is too high it will not flow, trapping the volatiles within the laminate, or if too low, then the resin will flow too much, creating areas of resin starvation. It is important that the temperature across the face of the mould tool remains relatively uniform to maintain uniform viscosity

of the resin. Therefore, Prudham [18] performed thermal imaging of the mould tool to gain insight into the temperature uniformity across the mould tool surface. Figure 9 shows the thermal imaging of the mould tool surface performed by Prudham [18].

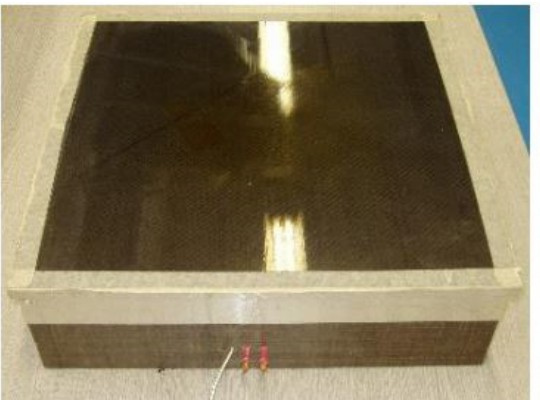
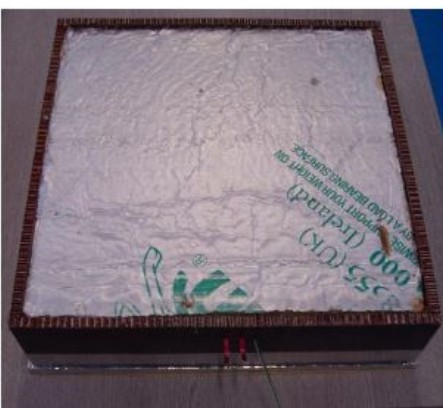

**Figure 8.** Heated mould tool used for cool-clave processing (acquired with permission from [18]).

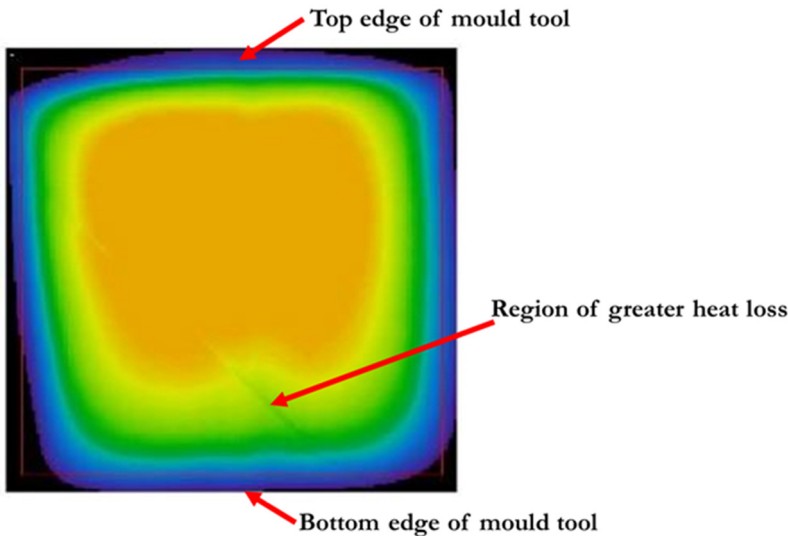

**Figure 9.** Thermal imaging of mould tool surface (acquired with permission from [18]).

The results indicated discrepancy between temperatures recorded by the mould tools built-in thermocouple and the temperatures recorded during thermal imaging technique. A greater heat loss near to the edges of the mould tool was recorded by thermal imaging with a maximum temperature difference across the main region of the mould tool being in the range from 5 °C horizontally and 7 °C vertically [18] in the image.

From the thermal imaging results, it was concluded that the mould tool would be suitable for cool-clave processing, demonstrating a convective heat transfer across the surface [18]. Greater heat loss across the edges of the mould tool surface was explained by in-plane anisotropic thermal conductivity of carbon fibre-reinforced epoxy composite used in the study [18,28]. The tests highlighted the need for adequate insulation of the mould tool during the manufacturing stage to prevent greater heat losses as recorded during thermal imaging [18].

A Thermal Insulation Hud (TIH) was constructed to prevent any convective heat loss from the mould tool which would encase the mould tool, allowing vacuum and thermocouple connections but limiting convective heat transfer [18]. One of the main challenges of the cool-clave technique was how to get power to the heated mould tool inside the autoclave. The electrical supply used to power the heated mould tool was provided by a power transformer controlled by an 'IMO' PID controller by using a pre-

existing access hole on the autoclave chamber, plugged together with a specially made fitting encompassing the electrical wires providing the pressure tight seal required [18]. A schematic of the power supply mechanism for the heated mould tool inside the autoclave is shown in Figure 10.

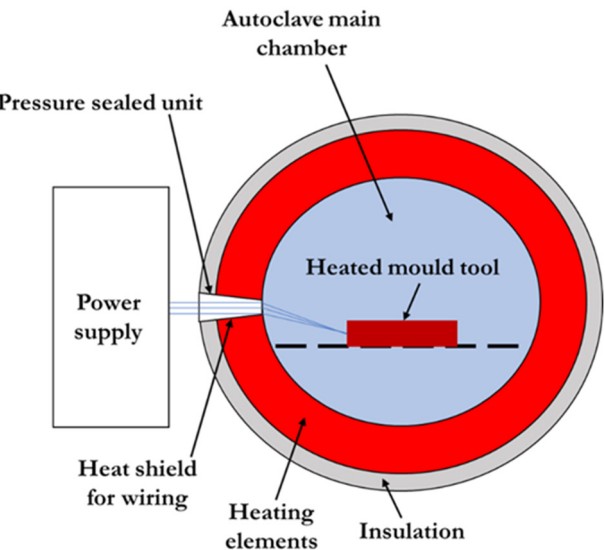

**Figure 10.** Schematic of the power-supply mechanism for the heated mould tool inside the autoclave chamber (redrawn from [18]).

### 3.1. Analysis of Energy Consumption between a Traditional Autoclave and Cool-Clave Technique

The main sources of energy consumption in a traditional autoclave and cool-clave process are compressor, and main controls (for both processes), heating elements in traditional autoclave, and heated mould tool in the cool-clave process. The principal objective behind cool-clave process is to achieve autoclave consolidation of composite laminates but being more energy- and cost-efficient. Prudham and Summerscales [27] quantified the energy consumption in a cool-clave processing technique during a defined cure cycle and compared it with a traditional autoclave process. The energy consumption by control units and compressor was deemed to be approximately equal for the traditional autoclave processing and cool-clave technique, as both processes require the autoclave to run with a pressurised chamber. To make a clear distinction between the energy consumption of both processes, the amount of energy required to provide heat during a defined curing cycle was evaluated [27]. The results demonstrated a 35% reduction in energy requirement for heating the laminate when replacing the vessel heaters with heated tooling [27].

The cure cycle in a trditional autoclave requires the component to enter the vessel at time 0 to start the dwell stage. Figure 11a illustrates a traditional autoclave cure cycle. Following thermodynamic analysis based on a traditional autoclave cure cycle, the total energy consumption reported by Prudham and Summerscales [27] was 3620 kJ. Table 1 further illustrates the results demonstrated in the study [27]. The cycle that an autoclave goes through to cure a laminate was split into six phases for the process of analysis. The analysis was carried out assuming the cure cycle of a 12.7 mm thick Cytec Cycom 5216 epoxy prepreg [27].

1.  heat from room temperature (20 °C) up to 80 °C
2.  pressurisation of autoclave at 80 °C to 0.7 MPa (gauge)
3.  reheat to 80 °C to compensate for the temperature drop during pressurisation
4.  hold temperature for 30 min
5.  heat from 80 °C up to 130 °C
6.  hold temperature for 95 min

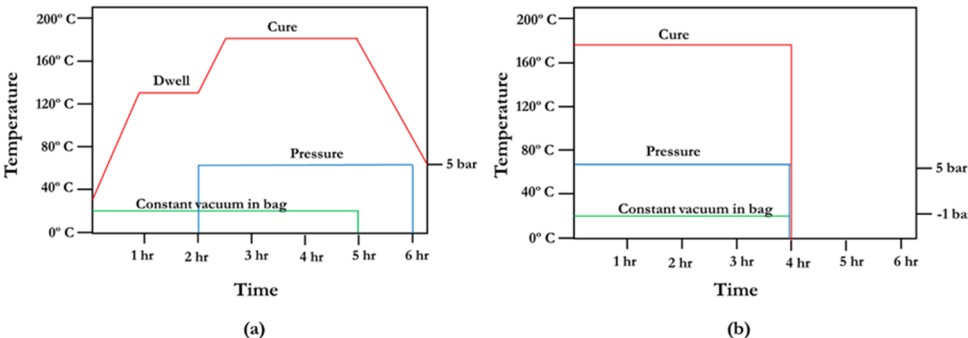

**Figure 11.** (**a**) Traditional autoclave cure cycle, (**b**) cure cycle for cool-clave processing (redrawn from [29]).

**Table 1.** Estimated energy consumption of a traditional autoclave cure cycle (reproduced from [12,27]).

| Phase | Start Temperature (°C) | Final Temperature (°C) | Energy Consumed (kJ) |
|---|---|---|---|
| 1 | 20 | 80 | 157.5 |
| 2 | 80 | 27.7 | 0 |
| 3 | 27.7 | 80 | 1061 |
| 4 | 80 | 80 | 306.4 |
| 5 | 80 | 130 | 1014.2 |
| 6 | 130 | 130 | 1081.2 |
| Total energy consumed | | | 3620.3 |

The total volume of pressurised gas inside the autoclave chamber was estimated to be 1 cubic metre through thermodynamic analysis [27], which was comprised of the cylindrical working space of the autoclave, annular heating channels, and domed ends.

Figure 11b illustrates the cure cycle for cool-clave processing technique [29]. In the proposed cool-clave cure cycle, the component can complete the dwell period and reach full heat prior to entering the vessel. This reduces the time required to be inside the vessel, thus increasing the production rate of components. Additionally, quality is key for acquiring the highest possible safety while minimising costs. Composites are excellent for tailoring for the loads and areas with stress concentrations and can provide tailorable tensile strength [29,30].

With the proposed method, the cycle times in industrial production line can be significantly reduced, which can bring a number of benefits, such as [29]:

1. load the autoclave after the dwell period
2. higher productivity
3. optimise profit potential
4. cure multiple components with different temperatures and cycle times
5. increase in the potential of new innovative products
6. less energy required for the process, a possibility of the use of clean energy

Following cool-clave processing using the heated mould tool technique, the maximum energy consumption based on processing of three Cytec Cycom 5216 560 gsm non-crimp glass fibre-reinforced epoxy composites with ~50% fibre volume fractions was estimated to be 2340 kJ, demonstrating a 35% reduction in energy consumption [12,27].

### 3.2. Consolidating Resin-Infused Laminates inside Autoclave

Lewin [31] conducted initial experiments as a brief feasibility study to investigate the possibility of consolidating resin-infused laminates inside an autoclave following RIFT II (Resin Infusion under Flexible Tooling) manufacturing outside the autoclave. Laminate (Plate E) properties were compared with laminates manufactured using hand lamination

with edge dams to constrain the flow of the infusion resin (Plate A), resin infusion under flexible tooling with a flow medium (RIFT II) and 0.03 MPa (Plate B), 0.06 MPa (Plate C), or 0.09 MPa (Plate D) net pressure [12,31]. Each laminate was manufactured using 270 gsm plain woven glass fabric-infused with IP2 polyester infusion resin (initial viscosity 1600 mPa.s at 25 °C, according to the manufacturer's datasheet) and 2% Butanox M50 MEKP catalyst by weight [12]. Laminate characterisation was performed by undertaking (a) resin burn-off for fibre volume fraction ($V_f$), (b) tensile properties (BS EN ISO 527-4), (c) flexural properties (BS EN ISO 14125 Class III), (d) inter-laminar shear strength (ILSS, BS EN ISO 14130), and (e) surface-breaking voids (SBV) by filling voids with carbon dust, followed by image processing and analysis with ImageJ software. The results are presented in Table 2.

**Table 2.** Results from each laminate characterization tests [31].

| Property | Units | Laminates | | | | |
|---|---|---|---|---|---|---|
| | | **A** | **B** | **C** | **D** | **E** |
| **Process** | | Hand lay-up | RIFT II | RIFT II | RIFT II | Autoclave |
| **Net pressure** | MPa | 0 | 0.03 | 0.06 | 0.09 | 0.586 |
| **Plate thickness** | mm | $2.34 \pm 0.06$ | $2.09 \pm 0.04$ | $2.03 \pm 0.05$ | $1.94 \pm 0.02$ | $1.93 \pm 0.03$ |
| $V_f$ **(thickness)** | % | 40.8 | 45.7 | 47.0 | 49.3 | 49.6 |
| $V_f$ **(burn-off)** | % | 42.0 | 46.5 | 46.7 | 48.1 | 50.9 |
| **Young's modulus** | GPa | $21.9 \pm 0.6$ | $23.7 \pm 0.3$ | $24.5 \pm 0.6$ | $24.9 \pm 0.4$ | $25.0 \pm 0.2$ |
| **Flex. modulus 40 mm span** | GPa | $18.7 \pm 0.4$ | $19.8 \pm 0.5$ | $19.8 \pm 0.5$ | $21.1 \pm 0.5$ | $22.4 \pm 0.4$ |
| **Flex. modulus 48 mm span** | GPa | $17.3 \pm 0.2$ | $19.0 \pm 0.3$ | $18.8 \pm 1.0$ | $20.4 \pm 0.3$ | $20.9 \pm 0.7$ |
| **Tensile strength** | MPa | $384 \pm 15$ | $416 \pm 20$ | $426 \pm 26$ | $452 \pm 31$ | $478 \pm 24$ |
| **Flexural strength** | MPa | $558 \pm 7$ | $578 \pm 13$ | $586 \pm 11$ | $599 \pm 21$ | $608 \pm 13$ |
| **ILSS 10.0 mm span** | MPa | $56.4 \pm 0.8$ | $53.5 \pm 1.3$ | $54.1 \pm 1.4$ | $53.4 \pm 1.6$ | $52.5 \pm 1.1$ |
| **ILSS 11.4 mm span** | MPa | $52.5 \pm 1.7$ | $49.9 \pm 1.7$ | $51.2 \pm 0.6$ | $46.6 \pm 0.9$ | $48.7 \pm 1.0$ |
| **SBV area** | % | 1.9 | 2.4 | 1.4 | 0.3 | 0.02 |

The fibre volume fraction, elastic moduli, and tensile and flexural strengths all increased with increasing net pressure during manufacturing. The ILSS decreased with increasing fibre volume fraction. The minimal gain in fibre volume fraction for Panel E was attributed to insufficient volume for resin bleed, with the possibility of a greater increase for an optimised process [12,31].

Stringer [32] identified 7500–16,500 mPa.s as an optimum processing window for application of vacuum for void-free high fibre volume fraction composites manufactured by wet lamination and vacuum bagging techniques. Therefore, with some adaptation of the autoclave pressure vessel, it might be practical to load the vacuum-bagged dry composite into the autoclave, then infuse and cure in situ. Preparation outside the autoclave would permit shorter autoclave cure cycles and better utilisation of the pressure vessel [12].

Improved Autoclave Process for Resin-Infused Laminates

Experiments conducted by Wilkinson [33] with a dry fabric reservoir inside the bag and with both the inlet and outlet pipes clamped gave no significant change in fibre volume

fraction (demonstrating only 0.26% increase) when compared to plates cured at ambient pressure. Subsequent tests used no reservoir material with the resin inlet clamped, while the resin outlet was vented to atmosphere during autoclave consolidation, demonstrating an 8.6% increase in fibre volume fraction [12,31,33].

After infusion, plates were subjected to (a) vacuum-bag only pressure, (b) 0.31 MPa pressure in the autoclave, or (c) 0.59 MPa pressure in the autoclave. Further laminates were prepared and pressurised after a dwell period to study the effect of viscosity at the time, and pressure was applied for the four times identified by the viscosity tests. The results are illustrated in Table 3.

**Table 3.** Summary of data obtained from laminates manufactured by outlet pipe vented to air [33].

| Property | Units | Infusion | 3.1/0.0 | 5.9/0.0 | 5.9/39.5 | 5.9/45 | 5.9/48.3 |
|---|---|---|---|---|---|---|---|
| Ext. pressure | MPa | 0.0 | $3.1 \times 10^{-4}$ | $5.9 \times 10^{-4}$ | $5.9 \times 10^{-4}$ | $5.9 \times 10^{-4}$ | $5.9 \times 10^{-4}$ |
| Dwell time | min | 0.0 | 0.0 | 0.0 | 39.5 | 45 | 48.3 |
| Plate thickness | μm | $2040 \pm 51$ | $1890 \pm 3$ | $1780 \pm 16$ | $1880 \pm 1$ | $1930 \pm 11$ | $1980 \pm 1$ |
| $V_f$ (thickness) | % | $51.9 \pm 1$ | $56.1 \pm 0.1$ | $59.3 \pm 0.5$ | $56.3 \pm 0.0$ | $54.7 \pm 0.3$ | $53.6 \pm 0.0$ |
| $V_f$ (burn-off) | % | 52.3 | 56.4 | 60.9 | 56.7 | 54.3 | 52.4 |
| Flexural modulus | GPa | $20.5 \pm 0.4$ | $25.6 \pm 0.9$ | $28.4 \pm 0.9$ | $26.4 \pm 0.7$ | $25.4 \pm 0.5$ | $24.0 \pm 1.0$ |
| Flexural strength | MPa | $347 \pm 14$ | $384 \pm 16$ | $415 \pm 20$ | $391 \pm 14$ | $383 \pm 10$ | $364 \pm 12$ |
| ILSS | MPa | $41.0 \pm 1.9$ | $44.5 \pm 2.5$ | $41.5 \pm 2.5$ | $42.5 \pm 3.4$ | $43.2 \pm 1.8$ | $42.4 \pm 1.7$ |

At constant consolidation pressure, delaying the consolidation resulted in a lower fibre volume fraction in the composite panel. Flexural strength increased with increasing consolidation pressure. Increased viscosity limited the quantity of resin expelled from the laminate, reduced the fibre volume fraction, and resulted in lower mechanical properties [33].

## 4. Conclusions

The out-of-autoclave manufacturing technique has limitations over maximum achievable fibre volume fraction in composites due to compressibility characteristics of the reinforcement material. Lower fibre volume fraction inevitably results in increased matrix volume fraction and consequent resin-rich volumes. Fibre clustering and RRV cause reductions in composite strength. Autoclave processing of composites is required to achieve the highest performance composites systems for industrial applications in the aerospace, automotive, and defence sectors.

Energy savings can result from decoupling the heat and pressure during the autoclave processing of composites. The use of electrically heated mould tools could eliminate heating of the parasitic systems (pressure vessel walls, insulation, and heat transfer gasses). The consolidation pressure can then be supplied using cool air.

The adoption of heated tooling in the cool-clave technique to achieve autoclave consolidation of composites can significantly reduce process cycle times, as the composites can be taken to the end of dwell period before loading the pressure vessel.

The use of resin infusion, rather than expensive pre-impregnated reinforcements, removes the need for the separate impregnation stage and elimination of power requirements for freezer storage of prepreg materials. Autoclave loading efficiency could be improved by curing different composite systems simultaneously with the composites brought to their respective curing temperatures before loading the autoclave, which would further enhance process efficiency.

**Author Contributions:** Conceptualization, J.S.; writing—original draft preparation, I.R.C.; writing—review and editing, I.R.C. and J.S. All authors have read and agreed to the published version of the manuscript.

**Funding:** This research received no external funding.

**Institutional Review Board Statement:** Not applicable.

**Informed Consent Statement:** Not applicable.

**Data Availability Statement:** No new data was created.

**Conflicts of Interest:** The authors declare no conflict of interest.

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
