# Peer review of "Cool-Clave—An Energy Efficient Autoclave"

_jcs, doi:10.3390/jcs7020082_

Round 1
Reviewer 1 Report
This manuscript mainly summarized the techniques to improve the energy and cost efficiency of the autoclave process using heated tools in the autoclave. The whole manuscript was well structured and informative, and was recommended to be accepted after minor revision.
1. It was recommended to emphasize the innovative nature of this study in the introduction.
2. Abstract, no references was allowed in this part.
3. The introduction part was a little weak, please strengthen this part.
4. P12, Line 416. The text description did not match Table 2, please verify.
Author Response
|
Reviewer 1 (Date of this review 31 Jan 2023 13:09:18) |
||
|
No. |
Comment |
Response |
|
1 |
This manuscript mainly summarized the techniques to improve the energy and cost efficiency of the autoclave process using heated tools in the autoclave. The whole manuscript was well structured and informative and was recommended to be accepted after minor revision. |
Thank you for the comment!
|
|
2 |
It was recommended to emphasize the innovative nature of this study in the introduction |
The innovative nature of this study is the cool clave technique. Relevant information regarding this technique has been emphasized in the introduction part in Page 2, from Line 55-69. We have discussed the technique in detail later in Section 3, therefore, we have decided to keep the information concise in the introduction part to avoid any repetition. |
|
3 |
Abstract, no references was allowed in this part |
Corrected in the manuscript |
|
4 |
The introduction part was a little weak, please strengthen this part |
Thank you for the comment! However, we have decided to keep the introduction part brief and concise. We have spoken briefly about Out-of-autoclave technique, autoclave consolidation and finally about the cool clave technique which are the focus of this paper. We feel the information provided in the introduction part is sufficient in context of the proposed paper and should motivate the reader to quickly focus on Sections 2 and 3 where the information is discussed in detail. Therefore, we would like to keep the introduction part of the paper unchanged. |
|
5 |
P12, Line 416. The text description did not match Table 2, please verify. |
Corrected in the manuscript |
Reviewer 2 Report
The article is well written. The scope is good and the coverage of the topic is appropriate.
A small detail that the authors may want to address is that in Section 2.1 (line 139) process pressure is given in MPa. Also, vacuum is given in Pa on line 250. However, in the remainder of the review article, the authors primarily discuss pressures in "bar/mbar". It may be beneficial to utilize a consistent unit of pressure, even though it is assumed that the units quoted are consistent with the specific citation be referred to.
Additionally, it seems that throughout the text of the manuscript, the unit "bar" is written without capitalization. However, in the steps to cure Cycom 5216 (line 367), the unit is given as "Bar".
Please check the manuscript for consistency.
Author Response
|
Reviewer 2 (Date of this review 07 Feb 2023 01:47:05) |
||
|
No. |
Comment |
Response |
|
1 |
The article is well written. The scope is good and the coverage of the topic is appropriate |
Thank you for the comment!
|
|
2 |
A small detail that the authors may want to address is that in Section 2.1 (line 139) process pressure is given in MPa. Also, vacuum is given in Pa on line 250. However, in the remainder of the review article, the authors primarily discuss pressures in "bar/mbar". It may be beneficial to utilize a consistent unit of pressure, even though it is assumed that the units quoted are consistent with the specific citation be referred to |
The pressure units are changed to MPa throughout the manuscript now. |
|
3 |
Additionally, it seems that throughout the text of the manuscript, the unit "bar" is written without capitalization. However, in the steps to cure Cycom 5216 (line 367), the unit is given as "Bar". Please check the manuscript for consistency. |
Checked and corrected in the manuscript |
Round 2
Reviewer 1 Report
This work is acceptable for publication.
Reviewer 2 Report
None.
Thank you for addressing the comments.